# Interventions to increase cervical screening uptake among immigrant women: A systematic review and meta-analysis

Zufishan Alam[1], Joanne Marie Cairns[2]*, Marissa Scott[3], Judith Ann Dean[4], Monika Janda[1]

1 Centre for Health Services Research, Faculty of Medicine, The University of Queensland, Brisbane, Queensland, Australia, 2 Hull York Medical School, University of Hull, Hull, United Kingdom, 3 School of Medicine, The University of Queensland, Brisbane, Queensland, Australia, 4 School of Public Health, Faculty of Medicine, The University of Queensland, Brisbane, Queensland, Australia

* joanne.cairns@hyms.ac.uk

**Data Availability Statement:** All relevant data are within the paper and its Supporting Information files.

## Abstract

Numerous intervention studies have attempted to increase cervical screening uptake among immigrant women, nonetheless their screening participation remains low. This systematic review and meta-analysis aimed to summarise the evidence on interventions to improve cervical screening among immigrant women globally and identify their effectiveness. Databases PubMed, EMBASE, Scopus, PsycINFO, ERIC, CINAHL and CENTRAL were systematically searched from inception to October 12, 2021, for intervention studies, including randomised and clinical controlled trials (RCT, CCT) and one and two group pre-post studies. Peer-reviewed studies involving immigrant and refugee women, in community and clinical settings, were eligible. Comparator interventions were usual or minimal care or attention control. Data extraction, quality appraisal and risk of bias were assessed by two authors independently using COVIDENCE software. Narrative synthesis of findings was carried out, with the main outcome measure defined as the cervical screening uptake rate difference pre- and post-intervention followed by random effects meta-analysis of trials and two group pre-post studies, using Comprehensive Meta-Analysis software, to calculate pooled rate ratios and adjustment for publication bias, where found. The protocol followed PRISMA guidelines and was registered prospectively with PROSPERO (CRD42020192341). 1,900 studies were identified, of which 42 (21 RCTS, 4 CCTs, and 16 pre-post studies) with 44,224 participants, were included in the systematic review, and 28 with 35,495 participants in the meta-analysis. Overall, the uptake difference rate for interventions ranged from -6.7 to 96%. Meta-analysis demonstrated a pooled rate ratio of 1.15 (95% CI 1.03–1.29), with high heterogeneity. Culturally sensitive, multicomponent interventions, using different modes of information delivery and self-sampling modality were most promising. Interventions led to at least 15% increase in cervical screening participation among immigrant women. Interventions designed to overcome logistical barriers and use multiple channels to communicate culturally appropriate health promotion messages are most effective at achieving cervical screening uptake among immigrant women.

**Funding:** The study was supported by Australian Government Research Training Program scholarship granted to ZA. JM is funded by Yorkshire Cancer research (Award reference number HEND405). The funders had no role in study design, data collection and analysis, decision to publish, or preparation of the manuscript.

**Competing interests:** The authors have declared that no competing interests exist.

## Introduction

Cervical cancer, despite being preventable, is a leading cause of cancer diagnosis and death among women worldwide, with 342,000 women dying in 2020 [1] Women in low and lower-middle income countries are most affected [2, 3]. Advances in biomedical research has led to the introduction of novel surgical, radiotherapeutic and systemic options for the treatment of cervical cancer [4]. Research evidence clearly shows that secondary prevention in terms of screening can effectively reduce cervical cancer mortality [5]. Screening options now being employed worldwide include Pap and HPV test [6]. Although many high-income countries have successful screening programs, disparities remain among certain population subgroups [7]. Immigrants have been identified as a subgroup with lower cervical screening uptake [8]. Therefore, multiple studies have delivered interventions to bring about better screening uptake among immigrant women globally.

Three systematic reviews have summarised studies involving health promotion interventions to increase cervical screening uptake among at-risk population subgroups. Of those, two focused on specific migrant groups i.e., Asian and Hispanic immigrant populations and indicated the role of sociocultural factors and population characteristics in intervention effectiveness [9, 10]. Whereas the third review on studies conducted between 2006–16 focused on activities for increasing cervical screening uptake among low socioeconomic groups, indicating effectiveness of HPV self-sampling [9]. Reviews have been carried out to summarise the evidence on interventions that used specific strategies such as education provision, Human Papilloma Virus (HPV) self-sampling or health care provider (HCP) counselling among the Indigenous/native women [11–14]. However, none of these previous reviews addressed the overall diverse immigrant populations in different parts of the world, nor summarised various intervention strategies for increasing cervical screening in immigrants. Given the recent launch of global initiative to eliminate cervical cancer as a public health problem by WHO [15], it is critical to systematically review the evidence on effectiveness of interventions, among under reached groups such as immigrants.

Thus, the objective of this study was to obtain the systematic evidence, expanding on immigrant population subgroups from various backgrounds, not limited to intervention strategies of specific type, as opposed to previous reviews and to compare the effect of intervention between intervention and control groups through meta-analysis. This review aimed to systematically summarise the global and up to date evidence on interventions aiming to increase cervical screening uptake among immigrant and refugee women, and quantify their effectiveness via providing a pooled estimate of the effect, through a meta-analysis. A further aim was to extract the characteristics of interventions most effective for increasing cervical screening uptake, in order to inform researchers and policy makers of the most promising intervention components to include in future interventions and to identify find any remaining knowledge gaps.

## Methods

This systematic review followed PRISMA 2020 guidelines [16] (checklist included in S1 Table). The protocol was registered with International Prospective Register of Systematic reviews (PROSPERO) Registration number: CRD42020192341. Refer to S1 File for published protocol.

### Study search

Pubmed, Scopus, EMBASE, CINAHL, PsycINFO, CENTRAL and ERIC were searched from inception to 12th October 2021. The search strategy was developed with guidance by a

professional librarian and combined the most appropriate keywords, MESH terms and Boolean operators, such as ((cervical cancer OR cancer of the cervix OR cervical neoplasm)) OR cervical cancer, uterine)) AND (((screening OR detection OR Pap test OR Pap smear)) OR cervical smear))) AND (((immigrant* OR migrant* OR refugee* OR emigrant*)) OR (emigrants and immigrants)))). S2 Table (a-h) provide the full electronic search strategy for each database. Additionally, bibliographies of included articles were hand-searched to identify other potentially relevant studies (S2 Table (i)). Titles and abstracts of studies were screened to identify interventions or health promotion activities aimed to increase cervical screening uptake among immigrant or refugee women. The database search was repeated in June 2022 to include any recently published studies.

## Study eligibility and selection

Original, peer-reviewed studies of any design ((randomised controlled trials (RCT), clinical controlled trials (CCT), cohort analytic pre-post (Quasi experimental) studies), with both simple and complex interventions were included, without restriction of language. Studies with interventions focusing only on increasing cervical cancer and screening knowledge, but not behaviour, and descriptive studies exploring patterns of cervical screening uptake among immigrant groups were excluded. Studies without complete outcome data were also excluded, after attempting to contact the authors for details. Studies involving immigrant and refugee women from any background were included. Conference proceedings and theses were excluded. Studies were independently retrieved and screened against inclusion criteria by at least two reviewers (ZA, JC, MS) via COVIDENCE, with resolution of any difference through mutual discussion.

## Data extraction

Fields predesignated by the authors were used to extract study data, including publication details (author, year), population characteristics (sample size, age, ethnicity, baseline screening status), study setting and location, recruitment method, intervention characteristics (type of intervention, control and intervention group, follow up period), and outcome measure(s). The outcome measure of primary interest for the systematic review was difference in cervical screening uptake from pre- to post-intervention in the intervention group, expressed as percentage. When the study reported more than one outcome measure, or calculations for different time intervals, the one with higher value was used. According to PRISMA guidelines, data were also extracted independently by at least two authors (ZA, MS, JC).

## Synthesis of extracted data and meta-analysis

Extracted data were then synthesised and reported narratively, arranging studies based on intervention type (simple/multifaceted), study setting (urban, rural community/clinical), source of outcome data measurement (self-reported/record based), screening method offered (self-sampling/pap test/ combined), mode of delivery (in person/via use of mail/telephone/media), intervention format (brochures/video/combined), guidance by a theoretical or behaviour change model (theoretically guided) and involvement of personnel (HCPs/Promotoras). Outcome data was reported with ranges across studies with similar characteristics.

The review was followed by meta-analysis of RCTs, CCTs and two-group pre-post studies. Meta-analysis was performed using Comprehensive Meta-Analysis (CMA) software Version 3 [17]. Due to wide variety of interventions used and populations addressed, random effects model was selected. The pooled effect size (ES) was calculated from the proportion of women screened post intervention in the intervention and control groups, respectively, and was

reported with 95% confidence intervals along with p values (p<0.05 considered as threshold for statistical significance). Q statistics and $I^2$ values were reported to inform about heterogeneity. A statistically significant Q value is indicative of heterogenous distribution of ES, whereas the $I^2$ statistic describes ES heterogeneity contributed by non-sampling error. Additionally, a prediction interval with 95% confidence interval was calculated, which is an accurate measurement of heterogeneity and variance of the ES, and gives more information on the distribution of effect than $I^2$ analysis alone [18]. To explore heterogeneity further, studies were then stratified into subgroups based on explanatory variables such as study type. Analysis was only performed when there were three or more studies available in a stratification group.

Publication bias was assessed by visual funnel plots inspection, assessment of symmetry via Egger's test and Begg–Mazumdar Kendall's Tau test. When bias was found, it was adjusted using trim and fill method introduced by Duval and Tweedie [19]. Sensitivity analysis was also conducted by removing studies with low quality (that scored weak on EPHPP scale), as well as an evident outlier with the highest effect size.

### Critical synthesis and quality appraisal of the studies

The quality of included studies was appraised using the Effective Public Health Care Practice Project (EPHPP) quantitative study quality assessment tool. This tool was first published in 1998 and effectively measures quality of intervention studies, especially in public health [20, 21]. It assesses six criteria: selection bias (representation by target population), study design and randomisation, confounders and their adjustment, blinding of participants and assessors, validity and reliability of data collection methods, and withdrawals and dropouts. The scores were determined by two independent reviewers (ZA, JC) and inter-rater reliability using Cohen's kappa calculated.

## Results

In total, 1,900 articles were retrieved from databases including Pubmed (392), Scopus (459), EMBASE (480), CINAHL (356), PsycINFO (140), CENTRAL (53) and ERIC (3) and bibliographies of the included articles (17) (Fig 1). After removal of 1,151 duplicates, 749 studies remained. Their titles and abstracts were searched to include relevant interventions, yielding 103 studies. Of these, 42 articles were chosen after a full text review, with 28/42 included in the meta-analysis. Remaining studies (61) were excluded as they lacked full text (10), had irrelevant outcomes/inadequate information on outcome measures (23), focused on increasing knowledge only (3), focused on intervention design (9) or included generalised information summary (7) only, addressed irrelevant populations (8), or consisted of review (1).

### Characteristics of the included studies

Of the 42 studies in total, 21 were RCTs, four CCTs, 12 (single group pre-post) and four (two group pre-post) cohort analytic studies. Table 1 provides the characteristics of overall studies included in the systematic review. The majority of studies (23) were conducted between 2011–2021. The number of participants ranged from 42–10,810, age ranged from 18–72 years, with similar participant characteristics in the controlled trials as in overall studies, while the cohort analytic studies had a smaller maximal number of participants (65–1,732).

Baseline screening status of participants in most of the studies (34/42) was under- or never-screened, however nine studies included participants who were up to date with screening as well. The majority of studies (36/42) were conducted in community settings (residences, churches, community centres, consulates); 31 in metropolitan and five in rural areas, whereas the rest (5/42) were conducted in healthcare settings (refugee/immigrant clinics). The majority

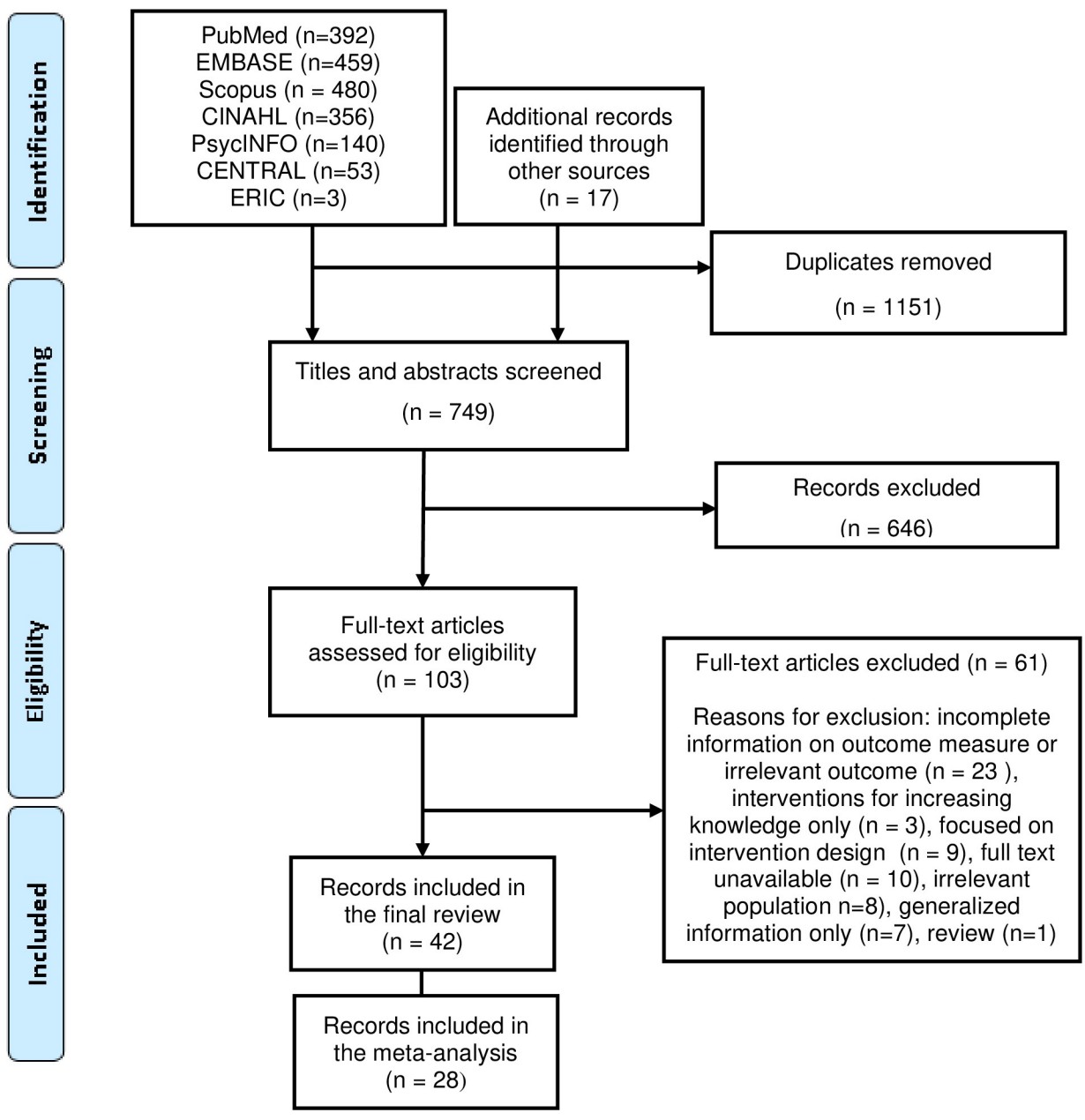

**Fig 1. PRISMA flowchart for the systematic review and meta-analysis process.**

of studies were conducted in the US (31), with relatively few in other countries: Canada (3), Hong Kong (3), UK (2), and Norway (2). Nearly one third (14) of the studies targeted multi-ethnic participants, whereas the rest involved immigrants from specific backgrounds only, including Latinas (4), Vietnamese (5), Hispanic (5), South Asian (4), Filipino (1), Chinese (2), Mexican (2), Korean (2), Cambodian (1), Somali (1) and Pacific Islander (1).

Most of the studies involved immigrant women from the community not belonging to any specific profession, while six studies focused on specific subgroups i.e., nail salon workers, farmworkers, and female sex workers (Table 2). The follow-up time after intervention ranged between two months to four years, with the majority having six months follow-up. Based on

**Table 1. Number of studies grouped by characteristics in the systematic review.**

| Study characteristics | Number of studies |
|---|---|
| **Screening outcome assessment** | |
| Self-reported | 25 |
| Record reported | 17 |
| **Complexity of intervention** | |
| Simple | 8 |
| Multifaceted | 34 |
| **Intervention components** | |
| With education only | 8 |
| With education and navigation | 3 |
| With education and reminders | 3 |
| With education in brochures only | 7 |
| With education in audiovisual help/media only | 13 |
| With education in brochures + Multimedia | 9 |
| Behavioural persuasion | 15 |
| Specific clinic involvement | 3 |
| With education, in combination with other aspects (reminders, navigation, financial incentive, behavioral techniques) | 25 |
| **Mode of delivery** | |
| In person | 36 |
| Mailed | 3 |
| Phone | 1 |
| Media | 1 |
| NS | 1 |
| **Setting** | |
| Urban Community | 31 |
| Rural | 5 |
| Clinical | 6 |
| **Location** | |
| United Kingdom | 2 |
| United States | 31 |
| Canada | 3 |
| Hong Kong | 4 |
| Norway | 2 |
| **Theoretically guided intervention** | |
| Yes | 29 |
| No | 13 |
| **Involvement of personnel in intervention delivery** | |
| Health Care Practitioners | 4 |
| Community Health Workers | 38 |
| **HPV self-sampling offered** | |
| Yes | 4 |
| No | 38 |
| **Date of publication** | |
| Before 2000 | 5 |
| 2001–2010 | 13 |
| 2011–2021 | 24 |
| **Study design** | |

*(Continued)*

**Table 1.** (Continued)

| Study characteristics | Number of studies |
|---|---|
| Randomised Control Trial | 20 |
| Clinical Controlled Trial | 4 |
| Cohort analytic (one group pre post study) | 14 |
| Cohort analytic (two group pre post study) | 4 |
| **Follow up period** | |
| Less than 6 months | 9 |
| 6 months | 17 |
| 7–12 months | 8 |
| More than 12 months | 5 |
| Not specified | 3 |
| **Study quality based on risk of bias** | |
| Strong | 5 |
| Moderate | 16 |
| Weak | 21 |

the EPHPP tool, most studies were weak in quality (21), followed by moderate (15) and strong (5).

## Intervention characteristics

Difference in cervical screening uptake ranged from 20–96% in the pre-post studies and -6.7 to 81% in controlled studies, for an overall range difference of -6.7 to 96% (Table 2). Almost all intervention studies focused on increasing cervical screening uptake through education, with eight using brochures or flip charts, 13 using audio-visual tools, and 10 using a combination of both. The screening uptake difference ranged from 16.7–81% for interventions using brochures, 2.4–87% for those using videos and -6.7 to 70% using a combination of both. The majority of the interventions (34/42) were delivered in person via Promotoras or health care workers, with three combining these with mailed materials [20, 28, 50], and one with media delivered education [31], whereas three solely used telephone, mail and media each [24, 29, 34]. Of the reviewed studies, 15 used behavioural intervention techniques beyond education such as motivation, persuasion and role modelling via survivors, celebrities, and narrative videos (screening uptake difference -6.7 to 77%). Three studies used specialised immigrant clinics to reach the target population (screening uptake difference 51−96%) [32, 55, 56].

Although most interventions promoted Pap test only, three focused on increasing HPV self-sampling in combination with Pap test, and resulted in increased cervical screening by 66−77%, compared to 11−48% increase in Pap test arms [42, 43, 47]. Another RCT offered self-sampling option only in person or by mail resulting increasing cervical screening by 81 and 72%, respectively [50]. Relatively few studies (3) involved health care practitioners in intervention delivery, of which one RCT, conducted in family doctor practices, yielded screening uptake increase by 2.6% [58] and two single group pre-post studies by 52−87% [40, 55]. Moreover, navigation, reminders and financial incentives as additional components of intervention were used in 26 studies, reporting screening uptake increases by 8−96%. Most studies (25) assessed screening uptake through self-reported uptake of -6.7 to81%, whereas 17 used objective measures such as medical record extraction reporting an increase of 2.6−96%. Not all controlled trials used completely unexposed control arms, seven studies used minimal intervention groups [33, 36, 41, 45, 47, 54, 61], three used intensive intervention groups as

**Table 2. Characteristics of the individual studies in the systematic review, with reported outcome measures.**

| Study reference | Location | Setting | Total Number of participants (intervention group, control group) | Ethnicity & age | Screening status at baseline (intervention group, control group) | Study design | Control group | Intervention Description | Intervention conduction time & Length of follow up | Source of outcome assessment data | Increase in cervical screening uptake post intervention (intervention group, control group) |
|---|---|---|---|---|---|---|---|---|---|---|---|
| Bird et al., 1998 [22] | US | Community neighbor-hoods, San Francisco | 717 345, 372 | Vietnamese women 18 years & above | 0%, 0% | RCT | Non exposed | Small group education sessions & educative material distribution via various venues | 1992–96 4 years | Self-reported | 20% intervention -3% control |
| McAvoy et al., 1991 [23] | UK | Community (Homes), Leicester | 737 216, 124 | Asian (15–52 years) | 0% 0% | CCT | Non exposed | Three arms versus control arm, 1) mailed information 2) visit by CHW with information, 3) visit by CHW with video | 1987–88 6 months | Medical records | 1) 11%, 2) 26%, 3) 30%) intervention 5% control |
| Kernohan et al., 1996 [24] | UK | Community venues, Bradford and Yorkshire | 1000 | Multi-ethnic (age NS) | 67% | Pre-post | NA | Educational sessions with audiovisual aid | 1991–93 6 months | Self-reported | 20% |
| Goldsmith at al., 1996 [25] | US | Rural farmworker community, California | 1732 | Hispanic Mexican (18 years & above) | 28% | Pre-post | NA | Multifaceted with several components | 1995–96, post test survey 45 days | Voucher redemption | 10.6% |
| Jenkins et al., 1999 [26] | US | 4 Counties (Homes), California | 876 422, 454 | Vietnamese 18 years & above | 54.4%, 43.6% | CCT | Non exposed | Media based intervention | 4 years 1992–96 | Self-reported | -6.7% intervention -6.6% control |
| Byrd et al., 2013 [27] | US | Community venues in El Paso, Houston and Yakima Valley. | 613 460, 153 | Farmworker Hispanic women 21 years & above | 0%, 0% | RCT | Non exposed | 3 arms (CHW visit with flipchart, CHW visit with video, CHW visit with flipchart & video) | 6 months 2008–09 | Self-reported | 41% flipchart, 45% video, 52% combined intervention 24.8% control |
| Jackson et al., 2002 [28] | US | Metropolitan areas of Seattle and Vancouver | 481 160, 161, 160, | Chinese 20–69 years | 0%, 0% | RCT | Non exposed | Two arms (Mailed educative material, CHW visit educative material) | 1999–2000 6 months | Self-reported | 25% mail 39% CHW visit intervention 15% control |
| Meade et al., 2002 [29] | US | Rural farmworker community, Florida | 65 | Hispanic farm worker women 18–72 years | 80% | Pre-Post | NA | Educational sessions with video, & offer of free tests at clinics | NS 6 weeks | Self-reported | 50% |

(Continued)

**Table 2.** (Continued)

| Study reference | Location | Setting | Total Number of participants (intervention group, control group) | Ethnicity & age | Screening status at baseline (intervention group, control group) | Study design | Control group | Intervention Description | Intervention conduction time & Length of follow up | Source of outcome assessment data | Increase in cervical screening uptake post intervention (intervention group, control group) |
|---|---|---|---|---|---|---|---|---|---|---|---|
| Taylor et al., 2002 [30] | US | Refugee community, Seattle | 370 196, 174 | Cambodian women aged 18 years & above | 44, 51% | RCT | Non exposed | Multifaceted with several components | 1997–98 12 months | Self-reported | 17% intervention 11% control |
| Jibaja-Weiss et al., 2003 [31] | US | Community health centres, Houston | 1574 984, 499 | Multi-ethnic (18–64) low income | 0%, 0% | RCT | Non exposed | Educational letters either personalised form/ or letter | NS 12 months | Medical records | 40% intervention 44% control |
| Maxwell et al., 2003 [32] | US | Community organisation, churches, Los Angeles | 530 213, 234 | Filipino women aged 40 years and above | 44%, 40% | RCT | Alternate intervention: Exercise | Information delivery with behavioral component | 1998–2000 12 months | Self-reported | 12% screening 12% control |
| Lam et al., 2003 [33] | US | Community venues, California | 400 200, 200 | Vietnamese American NS | 62.1%, 72.8% | RCT | Minimal intervention | Educational session by CHW | 2001–2002 2 months | Medical records | 14.8% intervention 2.6% control |
| Grewal et al., 2004 [34] | Canada | Community venues, British Columbia | NA (different at different time periods) | South Asian NS | 0% | Pre-Post study | NA | Pap test clinic for the women 2 days/week | 1995–2002 NA | Medical records | 50.6% |
| Black MEA et al., 2006 [35] | Canada | Community venues, Ontario | 500 | Multi-ethnic NS | 45% | Pre-post | NA | Educational sessions by CHW, navigational support | 4 months 2003–2005 | Self-reported | 38% |
| Dietrich et al., 2006 [36] | US | Community and migrant health centres, New York | 1413 696, 694 | Multi-ethnic women 50–69 years | 71% 70% | RCT | Minimal intervention | Tailored support through phone calls | 2001–04 18 months | Medical records | 8% intervention 0% control |
| Wong et al., 2008 [37] | Hong Kong | Outreach Clinic, Ziteng district | 245 female sex workers | Migrant sex workers aged 20–57 years | 0% | Pre-Post study | NA | Outreach Clinic twice a month, providing service to sex workers, information delivery, test at same site | 2004–05 NA; same day | Medical records | 96% |
| Fernandez et al., 2009 [38] | US | Community venues, Dallas, Los Angeles, North Orange County | 926 | Asian Pacific Islander women aged 18 years & above | 0% | Pre-Post study | NA | Encore plus intervention consisting of multiple components | 1996–98 6 months | Self-reported | 44.4% |
| O Brien et al., 2010 [39] | US | Community venues, Philadelphia | 120 60, 60 | Hispanic women 18–65 years | 47%, 48% | RCT | Non exposed | Informative sessions by Promotoras | NS 6 months | Chart review | 18% intervention 12% control |

*(Continued)*

**Table 2.** (Continued)

| Study reference | Location | Setting | Total Number of participants (intervention group, control group) | Ethnicity & age | Screening status at baseline (intervention group, control group) | Study design | Control group | Intervention Description | Intervention conduction time & Length of follow up | Source of outcome assessment data | Increase in cervical screening uptake post intervention (intervention group, control group) |
|---|---|---|---|---|---|---|---|---|---|---|---|
| Wang et al., 2010 [40] | US | Community organization, New York | 134 80, 54 | Chinese women aged 18 years & above | 0%, 0% | Pre-Post study | Minimal Intervention | Multifaceted with multiple components | NS 12 months | Medical records | 70% intervention 11% control |
| Nuno et al., 2011 [41] | US | Rural community, Arizona | 381 190, 191 | Hispanic women aged 50 years & above | 52%, 43% | RCT | Minimal Intervention | Multifaceted with multiple components | 2003–2006 2 years | Self-reported | 46% intervention 23% control |
| White et al., 2012 [42] | US | Community venues (churches) Birmingham, Alabama | 782 | Latina women aged 18 years & above | 40.9% | Pre-post | NA | Informative session by physicians, and survivors, free appointment, scheduling, childcare | Many years 2003–2009 NA | Medical records | 52.4% |
| Jandorf et al., 2014 [43] | US | Community venues, churches Arkansas, New York, Buffalo | 1752 1039, 713 | Latina 18 years & above | 53.1%, 55.1% | RCT | Alternate intervention; Diabetes education | Multifaceted with several components | 2007–09 8 months | Self-reported | 24% intervention 18.8% control |
| Sewali et al., 2015 [44] | US | Community Centre, Minnesota | 63 32, 31 | Somali women aged 30–70 years | 0%, 0% | RCT | Intensive intervention; Clinic Pap test | Two arms (clinic-based Pap test and home-based HPV self-sampling test) | 2013–14 3 months | Medical records | 65.5% self-sampling arm, 19.5% Pap test arm |
| Carrasquillo et al., 2015 [45] | US | Community venues, Miami Dade, Haiti and South Dade. | 601 207, 212, 182 | Hispanic, Haitian, non-Hispanic black women aged 30–65 years | 0%, 0% | RCT | Minimal intervention | 2 arms (Home based HPV test, clinic-based pap test) | (2011–14) 6 months | Self-reported | 77% HPV test 43% Pap test intervention 31% control |
| Han et al., 2015 [46] | US | Churches, Washington | 560 278, 282 | Korean women aged 21–65 years | 0% | CCT | Non exposed | Multifaceted with literacy focused education navigation and reminder | (2010–14) 6 months | Medical records | 54.5% intervention 9.2% control |
| Ma et al., 2015 [47] | US | Community venues, eastern regions of US | 1416 658, 758 | Vietnamese women 21–70 years | 0% | RCT | Minimal Intervention | Multifaceted | NS 12 months | Medical records | 60.1% intervention 1.6% control |
| Elder et al., 2016 [48] | US | Churches, San Diego | 436 219, 216 | Church going Latinas 18–65 years | 87.6%, 85% | CCT | Alternate exercise group | Multifaceted | 2011–14 12 months | Self-reported | 2.4% intervention 3% control |

*(Continued)*

**Table 2.** (Continued)

| Study reference | Location | Setting | Total Number of participants (intervention group, control group) | Ethnicity & age | Screening status at baseline (intervention group, control group) | Study design | Control group | Intervention Description | Intervention conduction time & Length of follow up | Source of outcome assessment data | Increase in cervical screening uptake post intervention (intervention group, control group) |
|---|---|---|---|---|---|---|---|---|---|---|---|
| Ilangovan et al., 2016 [49] | US | Clinical settings | 180 | Uninsured Haitian, Latina women aged 30–65 years | 0% | Pre-post study | NA | Self-sampling HPV test & pap, financial coverage | 2013–2014 5 months | Medical records | 67% HPV test, 33% Pap test |
| Luque et al., 2017 [50] | US | Rural farmworker community | 90 38, 52 | Haitian, Latina women 21–65 years | 0%, 0% | Pre-Post study | Non exposed | Small group educational delivery with audiovisual means | 2014–2015 6 months | Self-reported | 32% intervention 19% control |
| Dunn et al., 2017 [51] | Canada | Community venues Toronto | 1300 118, 344 | Multi-ethnic women aged 21–69 years | 0%, 0% | Pre-Post study | Non exposed | Multifaceted with multiple components | 2012–2013 2 years | Medical records | 26% intervention 9% control |
| Kobetz et al., 2018 [52] | US | Community venues South Florida | 600 300, 300 | Hispanic, Haitian & non-Hispanic black aged 30–65 years | 0%, 0% | RCT | Intensive intervention with mailed HPV kits | Multifaceted with multiple components delivered by CHW visit | 6 months | Self-reported | 81% intervention 72% control |
| Brown et al., 2018 [53] | US | Community venues, consulate | 421 | Hispanic women aged 18 years & above | 43% | Pre-post | NA | Consultation for health screening, Referral for free screening and telephone reminders | 2015–2016 6 months | Self-reported | 32% |
| Savas et al., 2018 [54] | US | Community venues El Paso, Texas | 627 314, 313 | Latina women aged 21 years & above | 0% 0% | RCT | Minimal intervention | Multifaceted with several components | | Self-reported | 41% intervention 44% control |
| Wong et al., 2019 [55] | Hong Kong | Community centres | 42 21, 21 | South Asian aged 25 years & above | 0% 0% | Pilot RCT | Non exposed | Multifaceted with several components | 3 months | Medical records | 28.5% intervention 23.5% control |
| Chan et al., 2019 [56] | Hong Kong | Community centres, NS | 371 | South Asian women 21 years & above | 0% | Pre-post | NA | Information provision via talk, video & booklet | 2016–17 One year | Self-reported | 40.7% |
| Kiser et al., 2020 [57] | US | Clinical setting, Tucson | 128 | Hispanic under insured aged 21–65 years | 0% | Pre-post | NA | Information delivery, with patient engagement, staff training, clinic case log | 2018 3 months | Medical records 87% | 87% |

*(Continued)*

Table 2. (Continued)

| Study reference | Location | Setting | Total Number of participants (intervention group, control group) | Ethnicity & age | Screening status at baseline (intervention group, control group) | Study design | Control group | Intervention Description | Intervention conduction time & Length of follow up | Source of outcome assessment data | Increase in cervical screening uptake post intervention (intervention group, control group) |
|---|---|---|---|---|---|---|---|---|---|---|---|
| Fernandez et al., 2020 [58] | US | Nail salons, Houston | 186 | Vietnamese nail salon workers aged 18 years & above | 63.5% | Pre-Post study | NA | Offer of education, brochures to share with family/friends & navigation | 2014–2017 5 months | Self-reported | 16.7% |
| Ochoa et al., 2020 [59] | US | Community venues, Los Angeles | 232 104, 128 | Mexican women aged 25–45 years | 21%, 3% | RCT | Intensive intervention: Non narrative film | Educative film (narrative) | 2013–2014 6 months | Self-reported | 38% narrative, 29% non-narrative |
| Moen et al., 2020 [60] | Norway | GP practice, 20 subdistricts of Bergen | 10360 5227, 5133 | Multi-ethnic women aged 25–69 years | 53%, 50.7% | RCT | Non exposed | Multifaceted with several components | 2017–2018 6 months | Medical records | 2.6% intervention 0.6% control |
| Wong 2021 [61] | Hong Kong | Community centres in various districts | 402 195, 192 | South Asian women aged 25 years & above | 0% | RCT | Non exposed | Multifaceted with several components | 2018–2020 3 months | Self-reported Control | 41.5% intervention 14.6% control |
| Choi 2021 [62] | US | Rural Community venues, Philadelphia | 51 25, 26 | South Korean women aged 21–65 years | 0%, 0% | Pre-Post study | Non exposed | Simple education session | 2017 3 months | Self-reported | 35% intervention 7.7% control |
| Qureshi 2021 [63] | Norway | Community venues near Oslo | 10810 1544, 9266 | Pakistani & Somali women 20–69 years | 45.9%, 44.1% | RCT | Non exposed | Simple educational session | 2017 6 months | Medical records | 4.9% intervention 1.4% control |

NA = not applicable, NS = not specified, CHW = community health worker, US = United States, UK = United Kingdom. CCT = clinical controlled trial, RCT = randomised controlled trials

control arms [44, 52, 59], whereas four used attention control interventions such as physical activity or diabetes education [30, 41, 46, 48]. When grouped based on type of control group, participants in the intervention group had a change in cervical screening of -6.6 to 24.8% in studies with non-exposed control groups, 8−77% with a minimally exposed control groups, 38−81% in intensive interventions control groups, and 3−19% in control groups offered attention control interventions.

## Results from meta-analysis

The 28 studies included in the meta-analysis had 35 intervention arms. They included 35,495 participants overall, 20,685 in the control arms and 14,810 in the intervention arms, respectively. Pooled ES of cervical screening yielded a rate ratio of 1.49 (95% confidence interval (CI): 1.36–1.65), (Fig 2) with a Q value of 402.2 and $I^2$ value of 93%, indicating high heterogeneity. Prediction interval calculation indicated that the true ES in 95% of the comparable populations would fall between 0.95–2.34, thus concluding that cervical screening intervention would likely, but not always, be effective in immigrant populations. However, funnel plot inspection (Fig 3), Begg–Mazumdar Kendall's Tau (–0.43, p = 0.001) and Egger tests

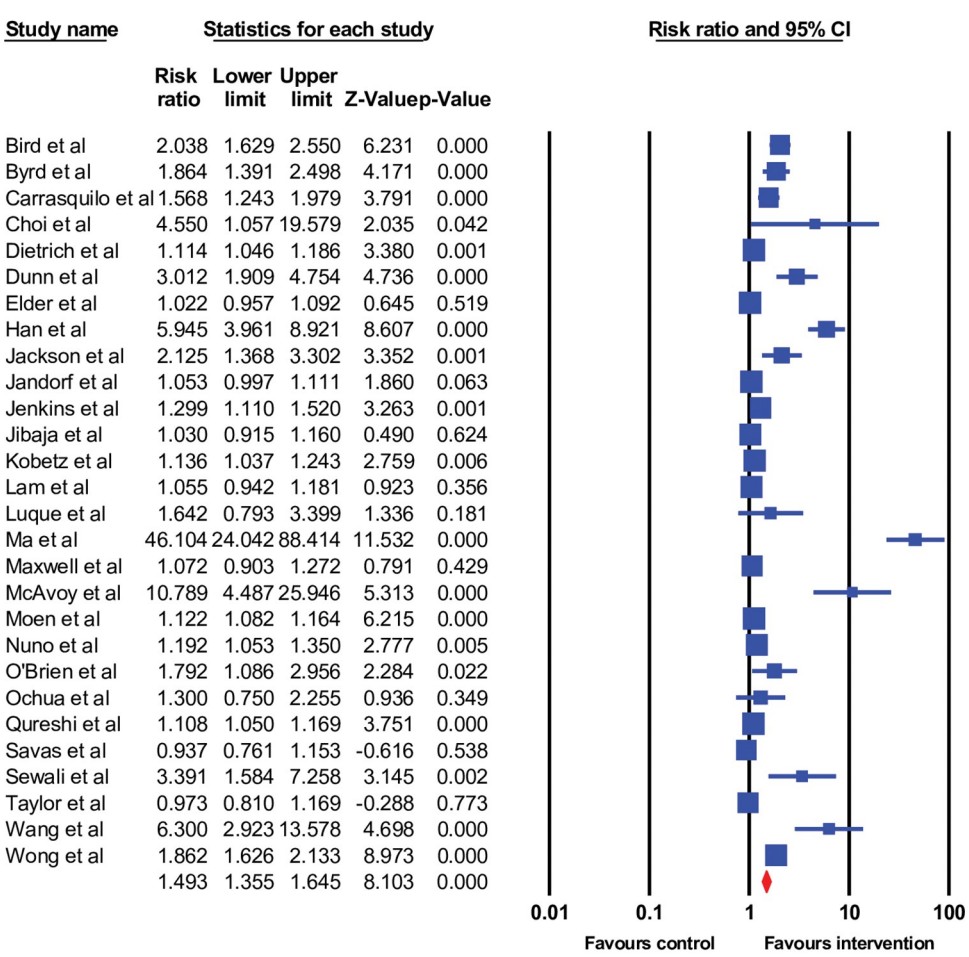

| Study name | Risk ratio | Lower limit | Upper limit | Z-Value | p-Value |
|---|---|---|---|---|---|
| Bird et al | 2.038 | 1.629 | 2.550 | 6.231 | 0.000 |
| Byrd et al | 1.864 | 1.391 | 2.498 | 4.171 | 0.000 |
| Carrasquilo et al | 1.568 | 1.243 | 1.979 | 3.791 | 0.000 |
| Choi et al | 4.550 | 1.057 | 19.579 | 2.035 | 0.042 |
| Dietrich et al | 1.114 | 1.046 | 1.186 | 3.380 | 0.001 |
| Dunn et al | 3.012 | 1.909 | 4.754 | 4.736 | 0.000 |
| Elder et al | 1.022 | 0.957 | 1.092 | 0.645 | 0.519 |
| Han et al | 5.945 | 3.961 | 8.921 | 8.607 | 0.000 |
| Jackson et al | 2.125 | 1.368 | 3.302 | 3.352 | 0.001 |
| Jandorf et al | 1.053 | 0.997 | 1.111 | 1.860 | 0.063 |
| Jenkins et al | 1.299 | 1.110 | 1.520 | 3.263 | 0.001 |
| Jibaja et al | 1.030 | 0.915 | 1.160 | 0.490 | 0.624 |
| Kobetz et al | 1.136 | 1.037 | 1.243 | 2.759 | 0.006 |
| Lam et al | 1.055 | 0.942 | 1.181 | 0.923 | 0.356 |
| Luque et al | 1.642 | 0.793 | 3.399 | 1.336 | 0.181 |
| Ma et al | 46.104 | 24.042 | 88.414 | 11.532 | 0.000 |
| Maxwell et al | 1.072 | 0.903 | 1.272 | 0.791 | 0.429 |
| McAvoy et al | 10.789 | 4.487 | 25.946 | 5.313 | 0.000 |
| Moen et al | 1.122 | 1.082 | 1.164 | 6.215 | 0.000 |
| Nuno et al | 1.192 | 1.053 | 1.350 | 2.777 | 0.005 |
| O'Brien et al | 1.792 | 1.086 | 2.956 | 2.284 | 0.022 |
| Ochua et al | 1.300 | 0.750 | 2.255 | 0.936 | 0.349 |
| Qureshi et al | 1.108 | 1.050 | 1.169 | 3.751 | 0.000 |
| Savas et al | 0.937 | 0.761 | 1.153 | -0.616 | 0.538 |
| Sewali et al | 3.391 | 1.584 | 7.258 | 3.145 | 0.002 |
| Taylor et al | 0.973 | 0.810 | 1.169 | -0.288 | 0.773 |
| Wang et al | 6.300 | 2.923 | 13.578 | 4.698 | 0.000 |
| Wong et al | 1.862 | 1.626 | 2.133 | 8.973 | 0.000 |
| | 1.493 | 1.355 | 1.645 | 8.103 | 0.000 |

**Fig 2. Forest plot for effect size (rate ratio) for getting screened post-intervention in overall 28 studies included in the meta-analysis.**

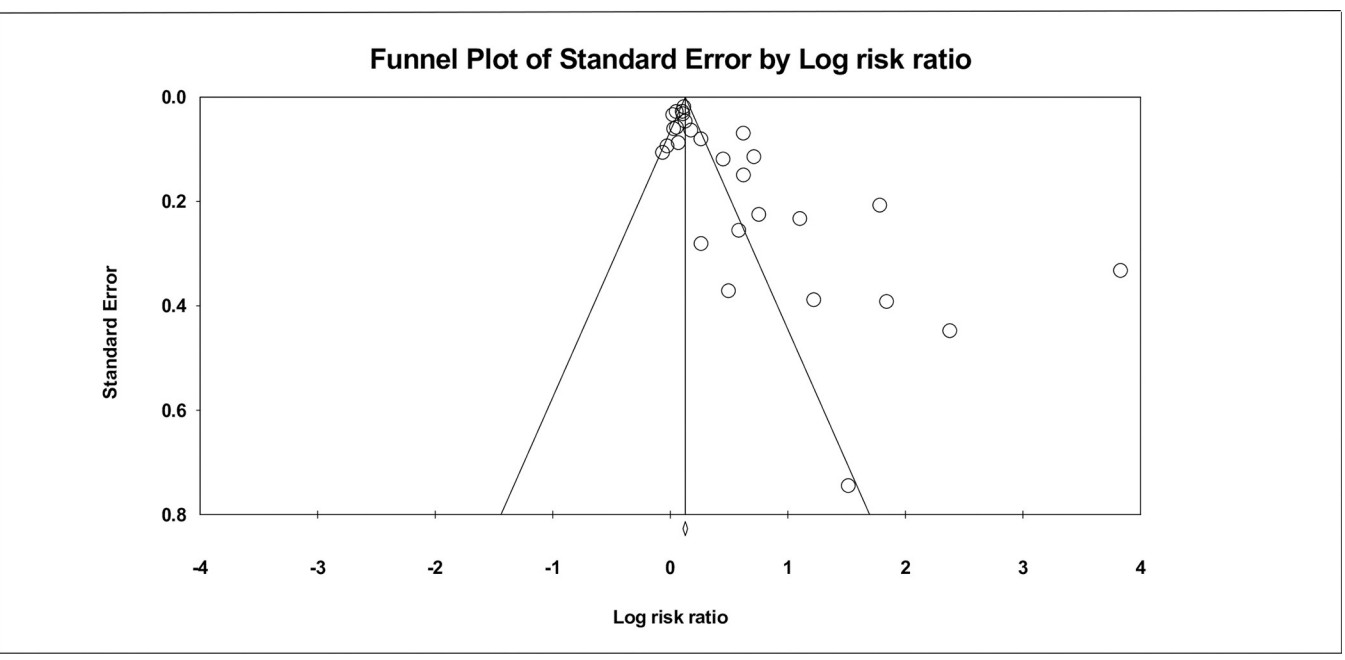

**Fig 3. Publication bias evident from the funnel plot for the overall studies included in the meta-analysis.**

(intercept = 3.66, p = 0.0001) indicated publication bias. Therefore, the ES was recalculated using Duval and Tweedie's trim and fill method, with 10 studies being adjusted, resulting in an ES of 1.15 (95% CI 1.03–1.29, p < 0.001) (Table 3). Substantial heterogeneity remained in almost all subgroups that were formed based on explanatory variables, indicating differences in intervention design, methodology and populations. Publication bias was also evident for most subgroups and adjusted accordingly (Table 3).

Subgroup meta-analysis indicated that when information and education was delivered using multiple modalities such as brochures and visual media strategies, ES (1.29 (95% CI: 0.83–2.00)) were higher, compared to using each modality alone (Refer to S1 Fig (a-k) for the subgroup forest plots). Results also suggested that multifaceted interventions had higher likelihood of increasing screening (ES 1.19 95% CI: (1.04–1.36)) compared to provision of education alone (ES 1.10 (95% CI: 0.97–1.24)). Studies with interventions delivered in person had a higher ES of 1.18 (95% CI: 1.04–1.35) compared to the ones that were not (ES 1.13 (95% CI 0.98–1.30)). It was observed that interventions with attention control arms such as exercise and diabetes education had lower effect (ES 1.04 (95% CI: 1.00–1.09)), compared to those with non-exposed control groups (ES 1.23 (95% CI: 1.04–1.44)) or minimal intervention groups (ES 1.15 (95% CI: 0.88–1.51)).

Interventions with under- or never-screened participants had considerably higher ES of 1.34 (95% CI: 1.00–1.81)), compared to those that also included participants up to date (ES 1.10 (95% CI: 1.06–1.14). Theoretically guided intervention studies had higher ES as did the ones involving community health workers and those conducted at broader level involving multiple locations within the country. No statistically significant difference in ES was seen in groups based on length of follow up, outcome source or study quality. When sensitivity analysis was conducted by removing studies with low quality [28, 63], or those evident as outliers in the funnel plot [47], heterogeneity indices remained the same and no effect was observed on the effect size.

**Table 3. Results for effect sizes among studies grouped by common variables, with observed heterogeneity and adjusted effect size for publication bias.**

| Study variable | No of studies included | Effect size (95% CI) | Heterogeneity | | | Trim & fill effect size (95% CI (adjusted studies) (No. of studies adjusted) | Classic fail-safe N |
|---|---|---|---|---|---|---|---|
| | | | Q | I² | p-value | | |
| All Studies [23, 26–28, 30, 31–33, 36, 39–41, 43–48, 50–52, 59–61] | 28 | 1.49 (1.36–1.65) | 402.2 | 93% | <0.001 | 1.15 (1.03–1.29) (10) | 2426 |
| **Study design** | | | | | | | |
| Controlled trials [22, 27, 28, 30, 32, 39, 43, 44, 46–48, 52, 54, 60, 61, 63] | 24 | 1.42 (1.29–1.56) | 8.21 | 93% | <0.001 | 1.15 (1.03–1.28) (7) | 1826 |
| Pre post (Quasi experimental) studies [40, 50, 51, 63] | 4 | 3.24 (1.85–5.65) | | | | 2.43 (1.29–4.27) (2) | 39 |
| **Outcome data source** | | | | | | | |
| Self-reported [22, 26–28, 30, 32, 36, 41, 43, 45, 48, 47, 60, 52, 54, 62] | 16 | 1.45 (1.13–1.68) | 1.02 | 93% | 0.3 | 1.11 (0.94–1.32) (6) | 502 |
| Cross validated by records [22, 31, 33, 36, 39, 40, 44, 46, 51, 60, 61, 63] | 12 | 1.62 (1.39–1.87) | | | | 1.26 (1.07–1.48) (4) | 706 |
| **Intervention complexity** | | | | | | | |
| Simple: Education alone [23, 28, 31, 33, 39, 63] | 6 | 1.12 (0.91–1.24) | 24.3 | 93% | <0.001 | 1.10 (0.97–1.24) (3) | 29 |
| Multifaceted: Education in combination with other strategies, such as reminder, navigation, financial incentive and behavioural motivation [22, 23, 26–28, 30, 32, 36, 40, 41, 43–48, 50, 51, 53, 54, 60–63] | 24 | 1.67 (1.48–1.88) | | | | 1.19 (1.04–1.36) (9) | 2164 |
| **Test type** | | | | | | | |
| Pap test [22, 23, 26–28, 30–33, 36, 39, 40, 43, 46–48, 50, 54, 59–63] | 25 | 1.50 (1.35–1.67) | 386.8 | 94% | <0.001 | 1.14 (1.01–1.28) (9) | 1964 |
| HPV test [52] | 1 | | NA | | | | |
| Pap + HPV test [44, 45] | 2 | | NA | | | | |
| **Length of follow up** | | | | | | | |
| 1 year or less [23, 24, 27, 28, 30–33, 36, 39, 40, 43–48, 52, 54, 60–63] | 22 | 1.51 (1.35–1.7) | 0.01 | 93% | 0.9 | 1.17 (1.03–1.34) (7) | 1457 |
| More than 1 year [22, 26, 36, 41, 50, 51] | 6 | 1.50 (1.20–1.88) 1.49 (1.21–1.84) | | | | 1.17 (0.93–1.47) (3) | 117 |
| **Mode of delivery** | | | | | | | |
| In person [22, 27, 28, 30, 32, 33, 39–41, 43–48, 50–52, 54, 60–63] | 23 | 1.69 (1.50–1.92) | 16.61 | 93% | <0.01 | 1.18 (1.04–1.34) (9) | 1985 |
| Mail/Telephone/Media [26, 28, 31, 36, 39] | 5 | 1.17 (1.03–1.33) | | | | 1.13 (0.98–1.30) 2 | 2 |
| **Mode of educative material** | | | | | | | |

*(Continued)*

**Table 3.** (Continued)

| Study variable | No of studies included | Effect size (95% CI) | Heterogeneity | | | Trim & fill effect size (95% CI (adjusted studies) (No. of studies adjusted) | Classic fail-safe N |
|---|---|---|---|---|---|---|---|
| | | | Q | I² | p-value | | |
| Brochure/Flipchart [23, 27, 31, 45, 48, 52] | 6 | 1.27 (1.06–1.49) | 11.69 | 94% | <0.01 | 1.18 (0.89–1.26) (3) | 52 |
| Video [23, 27, 28, 30, 36, 51, 63] | 7 | 1.70 (1.25–2.30) | | | | 1.16 (0.85–1.51) (3) | 97 |
| Brochure +Video [26, 27, 33, 40, 46, 47, 51, 54, 59] | 9 | 2.64 (1.74–4.01) | | | | 1.29 (0.83–2.00) (3) | 481 |
| **CHW involvement** | | | | | | | |
| Yes [22, 23, 26, 27, 28, 30, 32, 36, 39–41, 43–46, 48, 50, 52, 54, 60, 61, 63] | 23 | 1.67 (1.43–1.82) | 13.02 | 93% | <0.001 | 1.18 (1.02–1.35) (8) | 1890 |
| No [23, 26, 28, 31, 60, 62] | 6 | 1.19 (1.04–1.35) | | | | 1.13 () (0.97–1.32) (3) | 61 |
| **Type of control group** | | | | | | | |
| Usual/minimal [33, 36, 41, 44, 45, 47, 50, 52, 54, 61] | 10 | 1.69 (1.35–2.12) | 49.82 | 93% | <0.001 | 1.15 (0.88–1.51) (3) | 280 |
| Non exposed [22, 23, 26–28, 30, 31, 39, 46, 54, 51, 60, 61–63] | 14 | 1.70 (1.45–1.99) | | | | 1.23 (1.04–1.44) (6) | 897 |
| Attention control [32, 43, 48, 50] | 4 | 1.04 (1.00–1.09) | | | | Unchanged | 2 |
| **Baseline screening status** | | | | | | | |
| Not up to date at all [22, 23, 27, 28, 31, 40, 44–46, 47, 50, 51, 52, 54, 61, 62] | 15 | 2.62 (1.97–3.50) | 34.75 | 93% | <0.001 | 1.34 (1.00–1.81) (8) | 1280 |
| Mixed (up to date, not up to date) [26, 30, 32, 33, 36, 39, 41, 43, 48, 59, 60, 63] | 16 | 1.10 (1.06–1.14) | | | | Unchanged | 172 |
| **Theoretically guided** | | | | | | | |
| Yes [22, 27, 30, 31, 32, 33, 39–41, 43–48, 50, 51, 52, 54, 60–62] | 22 | 1.56 (1.38–1.77) | 402.1 | 93% | <0.001 | 1.18 (1.04–1.36) (7) | 1526 |
| No [22, 26, 28, 36, 59, 63] | 6 | 1.32 (1.13–1.56) | | | | 1.15 (0.95–1.38) (3) | 99 |
| **Location** | | | | | | | |
| Single [21, 28, 29, 30, 31, 34, 37, 38, 39, 42, 44, 46, 48, 49, 50, 51, 52, 57, 58, 60, 61] | 21 | 1.31 (1.20–1.44) | 402.2 | 93% | <0.001 | 1.15 (0.80–1.65) (4) | 349 |
| Multiple [24–26, 41, 43, 45, 59] | 7 | 2.26 (1.54–3.31) | | | | 1.14 (1.02–1.27) (6) | 921 |
| **Quality of studies** | | | | | | | |

(*Continued*)

**Table 3.** (Continued)

| Study variable | No of studies included | Effect size (95% CI) | Heterogeneity | | | Trim & fill effect size (95% CI (adjusted studies) (No. of studies adjusted) | Classic fail-safe N |
|---|---|---|---|---|---|---|---|
| | | | Q | I² | p-value | | |
| Strong [22, 26, 30, 36, 41] | 5 | 1.34 (1.08–1.65) | 4.69 | 93% | 0.1 | 1.10 (0.88–1.39) (2) | 58 |
| Moderate [27, 31, 32, 43, 44–47, 50, 52, 60–62] | 13 | 1.70 (1.43–2.01) | | | | 1.16 (0.95–1.41) (5) | 769 |
| Weak [23, 27, 33, 36, 39, 40, 48, 51, 53, 63] | 10 | 1.45 (1.23–1.73) | | | | 1.10 (4) (0.91–1.33) | 178 |

## Quality appraisal and risk of bias

Quality appraisal results suggested that the majority of the cohort pre-post studies were weak (13) due to lack of randomisation and participant blinding, contributing towards low overall quality scores. Common reasons for the controlled trials to be weak included selection bias due to lack of representative population when participants were conveniently sampled, or lack of details on confounder adjustment. Refer to S3 Table for description of individual quality criteria for all studies included in meta-analysis. Inter-rater reliability testing between the two quality raters (ZA, JC) yielded Cohen's Kappa of 0.4. The reason for low kappa score was differences in perception of rating criteria (selection bias, confounding) between the reviewers.

## Discussion

This study critically reviewed and meta-analysed interventions to increase cervical screening uptake among immigrant women globally. The review found that culturally appropriate interventions such as those providing HPV self-sampling methodology and targeted clinics for immigrants are most effective. Meta-analysis found that multicomponent interventions were more beneficial than single component ones, as were those theoretically guided, delivered in-person and using multiple formats of information delivery. Participant characteristics, such as baseline screening status also influenced the success of the interventions, as did the type of intervention selected for the control group.

To our knowledge this is the first systematic review and meta-analysis to comprehensively map the global evidence on the effectiveness of interventions to increase cervical screening in immigrant women. It compared the intervention effect sizes based on characteristics such as delivery format, involvement of HCPs, modes of educative material, screening status of participants and type of control group. The strengths of this study include following a prospectively registered protocol, clearly and transparently outlining our search strategy and methods of analysis, having multiple reviewers independently working on each review stage, and investigating novel methods of encouraging screening i.e., self-sampling, not included in previous intervention reviews on immigrants. However, this review also has certain limitations. First, although we conducted a comprehensive search of multiple databases, some studies might not have been located. We tried to overcome this through hand citation searches. Second, we found the EPHPP tool was more favorable towards experimental studies compared to non-controlled studies resulting in most studies being scored of weak quality. We also found significant heterogeneity in the studies limiting the number we could include in the sub-group meta-analyses.

The meta-analysis results suggested a low overall ES of 1.15 (95% CI 1.03–1.29) across all intervention studies after adjustment for publication bias. Although the results suggest a positive effect of interventions for cervical screening uptake, the results need to be interpreted with caution, in light of high heterogeneity ($I^2$ = 93%, Q = 402.2). However, heterogeneity often cannot be prevented in behaviour change studies, especially when dealing with diverse populations that require interventions adapted to their special needs. Similar heterogeneity has been observed in studies reporting meta-analysis of intervention studies for screening for other cancers [64]. Publication bias encountered in this review suggests that studies with less positive outcomes may exist but are difficult to trace [65]. Despite our efforts to find these unpublished studies, none could be discovered. Additionally, low Kappa score as evaluated in our study indicates weak interrater reliability on the quality appraisal of the studies. However, the literature suggests that the Kappa index may amplify disagreement estimate among the raters [66]. A low Kappa index is more of a concern when dealing with diagnostic tests in clinical studies [67] compared to quality appraisal, as in the current study.

The interventions included in the systematic review ranged from simple approaches such as mere provision of information, to those incorporating multiple components such as support of women using behaviour change techniques, patient navigation and practical help (provision of clinics for immigrants, childcare and transport). It was evident that the complex multifaceted interventional options, addressing broad areas of behavioural change and helping overcome the logistic constraints, were more effective at improving cervical screening uptake. It is similar to what has been reported for screening uptake for other cancers [64, 68], and for cervical screening among women generally [9]. In contrast to the systematic review findings, meta-analysis suggested that combined modes of information provision such as brochures, visual media as well as written information are more effective than using each of these strategies alone, similar to previous findings [69]. The difference between systematic review and meta-analysis findings could reflect that the meta-analysis mainly included higher quality studies and trials. Interventions that were guided by theoretical behaviour change models also had statistically significant stronger ES compared to the studies which were not. The advantages offered by interventional designs based on theoretical models have been summarised previously [70].

Interventions in broad populations, including both under- and well-screened women, compared to those including under- or never-screened women only, were less effective. Previous research also reported better effectiveness of risk-targeted rather than population-based interventions [71]. Thus, choosing the population on which to intervene is important, although it might be less convenient to reach a specific proportion rather than an entire immigrant population subgroup.

One of the intervention methods that recently became available and seems promising is HPV self-sampling. Although meta-analysis could not be performed due to the limited number of self-sampling studies available, the systematic review reported it resulted in higher screening compared to other interventions. A previous systematic review and meta-analysis reported self-sampling is more effective in increasing screening participation than traditional Pap testing in women generally [14]. Various barriers to screening common among immigrant women, such as modesty, religious reasons, and female HCP preference favour usage of HPV self-sampling in this population [72]. Similarly, HCP involvement significantly improved screening uptake, although few studies of interventions targeting HCP behaviours have been carried out. Interventions aimed at HCPs alongside the women could be important in enhancing screening uptake as suggested for other cancers [73]. The systematic review also reported the advantage of use of specialised clinics to reach immigrant women and enhance their screening uptake. Although meta-analysis could not be performed due to lack of eligible

studies, a previous meta-analysis indicated specialised clinics to be strongly effective in increasing immunisation and cancer screening uptake among adults [74]. Likewise, cultural appropriateness is important when dealing with individuals of immigrant background, with availability of materials in the women's native language being critical. All studies in this review used the same languages as that of the participants and often involved lay community health workers, such as *Promotoras*; this personalised approach was effective in the meta-analysis. Therefore, policies designed to address cervical screening services and interventions for migrants should take into consideration relevance of cultural responsiveness when including components such as information provision, navigation as well as financial incentives.

Another interesting finding from the systematic review was higher screening uptake rates when outcomes were assessed objectively rather than through self-report, although meta-analysis did not report statistically significant difference among the two groups, A possible explanation could be greater reliability of clinical records which can be obtained without the need of follow-up of each individual participant, reducing the non-response bias.

None of the included study reported cost-effectiveness of the intervention used, it is recommended to include this outcome in future studies. This would be of high practical value, allowing the policymakers to understand the choice of intervention, including of HPV self-sampling method. A recent systematic review of studies assessing cost of HPV self-sampling compared to standard screening strategies, reported it to be highly cost-effective for under screened women in high income countries, either when offered alone or in combination with other strategies [75]. Furthermore, effect of certain variable of interest, age and such as length of stay of participants in the country, could be explored through meta-regression in future studies.

In conclusion, this review identified a large number of studies, that have evaluated interventions to increase uptake of cervical screening amongst immigrant women. The findings suggest that interventions with multifaceted, culturally sensitive components, addressing practical challenges and including HPV self-sampling modality, could lead to significant increase in cervical screening participation among immigrant women. Review findings also suggest that using multiple channels to communicate with the target audience is the next most important feature of a likely successful intervention strategy. However, due to substantial heterogeneity observed in the meta-analysis results, intervention effects need to be interpreted cautiously. There is opportunity to study interventions that involve trusted HCPs [76, 77]. We recommend future research on this topic adopts robust study designs to improve the quality of the studies and avoid potential contamination. Theoretically guided interventions, targeted in their approach to ensure recruitment of women who could benefit most from an intervention, are recommended.

## Supporting information

**S1 File. Published PROSPERO protocol for the systematic review and meta-analysis.**
(PDF)

**S1 Fig. Forest plots showing sub-group analysis for cervical screening intervention studies included in the meta-analysis.**
(DOCX)

**S1 Table. PRISMA 2020 checklist as used in the review process.**
(DOCX)

**S2 Table. Search strategy as used in different databases.**
(DOCX)

**S3 Table. Quality appraisal of studies included in the systematic review.**
(DOCX)

## Author Contributions

**Conceptualization:** Zufishan Alam, Judith Ann Dean, Monika Janda.

**Data curation:** Zufishan Alam, Joanne Marie Cairns, Marissa Scott.

**Formal analysis:** Zufishan Alam, Monika Janda.

**Investigation:** Zufishan Alam, Joanne Marie Cairns, Marissa Scott.

**Methodology:** Zufishan Alam, Monika Janda.

**Software:** Zufishan Alam.

**Supervision:** Judith Ann Dean, Monika Janda.

**Visualization:** Zufishan Alam.

**Writing – original draft:** Zufishan Alam.

**Writing – review & editing:** Zufishan Alam, Joanne Marie Cairns, Judith Ann Dean, Monika Janda.

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
