## [Decision Letter · Decision Letter 0]

8 Nov 2022

PONE-D-22-22346Interventions to increase cervical screening uptake among immigrant women: a systematic review and meta-analysis

PLOS ONE

Dear Dr. Zufishan Alam,

Thank you for submitting your manuscript to PLOS ONE. After careful consideration, we feel that it has merit but does not fully meet PLOS ONE’s publication criteria as it currently stands. Therefore, we invite you to submit a revised version of the manuscript that addresses the points raised during the review process.

We look forward to receiving your revised manuscript.

Kind regards,

Gulzhanat Aimagambetova

Academic Editor

PLOS ONE

Journal Requirements:

2. We note that you have referenced (ie. Bewick et al. [5]) which has currently not yet been accepted for publication. Please remove this from your References and amend this to state in the body of your manuscript: (ie “Bewick et al. [Unpublished]”) as detailed online in our guide for authors

Reviewers' comments:

Reviewer's Responses to Questions

**Comments to the Author**

1. Is the manuscript technically sound, and do the data support the conclusions?

Reviewer #1: Yes

Reviewer #2: Yes

2. Has the statistical analysis been performed appropriately and rigorously? 

Reviewer #1: Yes

Reviewer #2: Yes

3. Have the authors made all data underlying the findings in their manuscript fully available?

Reviewer #1: Yes

Reviewer #2: Yes

4. Is the manuscript presented in an intelligible fashion and written in standard English?

Reviewer #1: Yes

Reviewer #2: Yes

5. Review Comments to the Author

Reviewer #1: This is a very well presented work, and it is methodologically correct. It is easy to follow and the conclusions are appropriate. From my point of view, it meets the criteria to be published. However, I would like to raise some issues or corrections to the authors.

1. On page 4, line 88, it is commented that the references of the included articles were revised. It would be good if the results were commented on which result this search brought. I think a point is made only in the discussion.

2. In table 1, the sum should always be 42. Could you please check if this is an error or if this is data not available? In this case, I recommend adding it to the table.

3. On page 9, line 183, specific subgroups workers are discussed, but it is not clear to which other groups the rest belong and whether the data is in the table or not

4. The authors show an heterogeneity index of 93%, and add: "indicating considerable heterogeneity". From my point of view, it would be a matter of high heterogeneity. In any case, it would be better not to make any assessment in the results section and deepen the impact of the same on the possible conclusions.

5. On page 28, first sentence, the authors comment about a Kappa index of 0.4. Again, I recommend to comment on this in the discussion section.

6. In the discussion, line??, the authors comment that the meta-analyses suggested a moderate ES of 1.15 (95% CI 1.03-1.29). I consider that it could be considered a low ES, taking into account that the lower limit is 1.03, and the upper one is 1.29.

7. Finally, I suggest the authors to include some discussion or consideration about the high cost-benefit of the interventions, given the small benefit shown by the pooled analyses, and the type of intervention (targeted at specific groups) that might seem more effective. How much money to save one woman from death from cancer de cèrvix, or how much to detect one cervical cancer?

8. There seems to be a mistake in reference 67. Could you please ckeck?

Reviewer #2: 1. Generally, I think the Introduction section is written well. However, it would be better if the authors can strengthen the following, as well as in Discussion: (1) compared to previous “individual” studies (i.e., not a meta-analysis level study) in this topic, what is the new value from the meta-analysis would be expected to bring into? (2) why executing this meta-analysis is so crucial for this topic? The considerations should greatly beyond “None of these previous reviews quantified the extent of benefit through meta-analysis” (page 3, lines 67-68); and (3) what are the questions this meta-analysis can further address but have not been answered in the past?

2. The authors mentioned “Sensitivity analysis was also conducted…” (page 6, lines 139-140). I would suggest indicating clearly here about what a study with low quality exactly was (e.g., giving clear definition and calculation on quality score, an instance as well).

3. In Table 1 (page 7), why for the characteristic “Complexity of intervention”, the total number is not 42 instead of 41 (8+33)?

4. In Table 1 (page 8), would it be better if adjusting “Country” to “Location”? It seems that not all candidates are countries.

5. Followed by the previous one, please comment/judge that would it be an issue/effect on the meta-analysis result if pooling studies altogether but their location units were not the same (e.g., some from country level and others from city level)?

6. Can the authors add some descriptions ahead (e.g., in the Method section) or somewhere appropriate about what the characteristic “Theoretically guided intervention” exactly indicate?

7. Would there be any reason that the characteristic “Date of publication” divide into the three categories? It would be better for readers if clarifying and adding this on the manuscript.

8. In Discussion, please give a more deep discussion on the following: (1) future research implication and direction based on this meta-analysis; and (2) relevant screening policies implications for immigrants.

6. PLOS authors have the option to publish the peer review history of their article (what does this mean?). If published, this will include your full peer review and any attached files.

Reviewer #1: **Yes: **Dr Marisa Baré

Reviewer #2: No

---

## [Author Response · Author response to Decision Letter 0]

7 Dec 2022

Response to reviewers

We are grateful to both the reviewers for providing us feedback and opportunity to improve the manuscript. We have endeavoured to revise the manuscript thoroughly in line with the suggestions. We have provided detailed explanation to each comment raised by the reviewer, the response is given (in blue) and changes/additions in the manuscript text (in red) (response to reviewers document).

Reviewer #1: This is a very well presented work, and it is methodologically correct. It is easy to follow and the conclusions are appropriate. From my point of view, it meets the criteria to be published. However, I would like to raise some issues or corrections to the authors.

1. On page 4, line 88, it is commented that the references of the included articles were revised. It would be good if the results were commented on which result this search brought. I think a point is made only in the discussion.

The references for studies obtained from bibliography of relevant articles have been added as an additional supplemental table that has been referenced in the text to show the results brought by this research, as follows:

Additionally, bibliographies of included articles were hand-searched to identify other potentially relevant studies (S3 table i) (Page 5, Line 111-112)

2. In table 1, the sum should always be 42. Could you please check if this is an error or if this is data not available? In this case, I recommend adding it to the table.

Thank you for bringing this to our attention, the number of studies in each category in Table 1 has been revised accordingly.

3. On page 9, line 183, specific subgroups workers are discussed, but it is not clear to which other groups the rest belong and whether the data is in the table or not

Other groups refer to general women in community, not belonging to any specific profession. Table 2 gives the information on ethnicity, and specifies if any special group was present (Nail salon workers, farmworkers and migrant sex workers). Further elaboration has been provided in text as follows:

Most of the studies involved immigrant women from the community not belonging to any specific profession, while six studies focused on specific subgroups i.e., nail salon workers, farmworkers or female sex workers (Table 2) (Page 10, Lines: 213-2015)

4. The authors show a heterogeneity index of 93%, and add: "indicating considerable heterogeneity". From my point of view, it would be a matter of high heterogeneity. In any case, it would be better not to make any assessment in the results section and deepen the impact of the same on the possible conclusions.

We agree with your comment, the text in results has been revised to replace considerable with high (line 43). Conclusion has been revised to add:

However, due to substantial heterogeneity observed in the meta-analysis results, intervention effects need to be interpreted cautiously. (Page 32, Lines: 422-424)

5. On page 28, first sentence, the authors comment about a Kappa index of 0.4. Again, I recommend to comment on this in the discussion section.

Comment regarding Kappa index has been added in the discussion section as follows:

Additionally, low Kappa score as evaluated in our study indicates weak interrater reliability on the quality appraisal of the studies. However, the literature suggests that the Kappa index may amplify disagreement estimate among the raters [64]. A low Kappa index is more of a concern when dealing with diagnostic tests in clinical studies [65] compared to quality appraisal as in the current study. (Page 29, Lines: 336-340)

6. In the discussion, line??, the authors comment that the meta-analyses suggested a moderate ES of 1.15 (95% CI 1.03-1.29). I consider that it could be considered a low ES, taking into account that the lower limit is 1.03, and the upper one is 1.29.

Agreed, we have replaced ‘moderate’ with ‘low’ ES. (Page 29, Line 327)

7. Finally, I suggest the authors to include some discussion or consideration about the high cost-benefit of the interventions, given the small benefit shown by the pooled analyses, and the type of intervention (targeted at specific groups) that might seem more effective. How much money to save one woman from death from cancer de cervix, or how much to detect one cervical cancer?

We are grateful for bringing to our attention this point, it has been added as suggested in the discussion as follows:

None of the included studies reported cost-effectiveness of the intervention used, it is recommended to include this outcome in future studies. This would be of high practical value, allowing the policymakers to understand the value of the choice of intervention, including of HPV self-sampling method. A recent systematic review of studies assessing cost of HPV self-sampling compared to standard screening strategies, reported it to be highly cost -effective for under-screened women in high income countries, either when offered alone or in combination with other strategies. (Page 30, Lines 391-397)

8. There seems to be a mistake in reference 67. Could you please check? 

Reference has been revised as follows:

Eccles M. The Improved Clinical Effectiveness Through Behavioral Research Group (ICEBERG). Designing theoretically-informed implementation interventions. Implementation Science. 2006;23;1(1):4.

Reviewer #2: 1. Generally, I think the Introduction section is written well. However, it would be better if the authors can strengthen the following, as well as in Discussion: (1) compared to previous “individual” studies (i.e., not a meta-analysis level study) in this topic, what is the new value from the meta-analysis would be expected to bring into? (2) why executing this meta-analysis is so crucial for this topic? The considerations should greatly beyond “None of these previous reviews quantified the extent of benefit through meta-analysis” (page 3, lines 67-68); and (3) what are the questions this meta-analysis can further address but have not been answered in the past?

We thank reviewer two for the suggestion on further elaboration on the objectives of the meta- analysis. Introduction and discussion sections have been revised to include the suggested description:

Introduction

Three systematic reviews have summarised studies involving health promotion interventions to increase cervical screening uptake among at-risk population subgroups. Of those, two focused on specific migrant groups i.e., Asian and Hispanic immigrant populations and outlined the significance of of sociocultural factors and population characteristics in intervention effectiveness [7,8]. Whereas the third review of studies conducted between 2006-16 focused on activities for increasing cervical screening uptake among low socioeconomic groups indicating effectiveness of HPV self-sampling [9]. Reviews have been carried out to summarize the evidence on interventions that used specific strategies such as education provision, Human Papilloma Virus (HPV) self-sampling or health care provider (HCP) counselling, among the Indigenous/native women [10-12]. However, none of these previous reviews addressed the overall diverse immigrant populations in different parts of the world, nor summarized various intervention strategies for increasing cervical screening in immigrants. Given the recent launch of global initiative to eliminate cervical cancer as a public health problem by WHO [13], it is critical to systematically review the evidence on effectiveness of interventions, among under reached groups such as immigrants. 

Thus, the objective of this study was to obtain the systematic evidence, expanding on immigrant population subgroups from various backgrounds, not limited to intervention strategies of specific type, as opposed to previous reviews and to compare the effect of intervention between intervention and control groups through meta-analysis. This review aimed to systematically summarise the global and up to date evidence on interventions aiming to increase cervical screening uptake among immigrant and refugee women, and quantify their effectiveness via providing a pooled estimate of the effect, through a meta-analysis. (Page 3, Lines 60-63, Page 4, Lines 70-72)

Discussion

Therefore, to our knowledge this is the first systematic review and meta-analysis to comprehensively map the global evidence on the effectiveness of interventions to increase cervical screening in immigrant women. It compared the intervention effect sizes based on characteristics such as delivery format, involvement of HCPs, forms of educative material, screening status of participants and type of control group. (Page 30, Lines 410-415)

2. The authors mentioned “Sensitivity analysis was also conducted…” (page 6, lines 139-140). I would suggest indicating clearly here about what a study with low quality exactly was (e.g., giving clear definition and calculation on quality score, an instance as well).

Further explanation has been included as follows in the methods section whweras the example of study has been given in results section:

Sensitivity analysis was also conducted by removing studies with low quality (that scored weak on EPHPP scale) as well as an evident outlier with the highest effect size. (Page 7, Lines 166-167, Methods section)

When sensitivity analysis was conducted by removing studies with low quality [26,61],, or those evident as outliers in the funnel plot [45], heterogeneity indices remained the same, and no effect was observed on the effect size. (Page 26, Line 306, Results section)

3. In Table 1 (page 7), why for the characteristic “Complexity of intervention”, the total number is not 42 instead of 41 (8+33)?

The number has been revised and corrected.

4. In Table 1 (page 8), would it be better if adjusting “Country” to “Location”? It seems that not all candidates are countries.

We agree, term ‘country’ has been replaced with ‘location’.

5. Followed by the previous one, please comment/judge that would it be an issue/effect on the meta-analysis result if pooling studies altogether but their location units were not the same (e.g., some from country level and others from city level)?

Further analysis of studies conducted at multiple locations (country level) and single location (city level), has been included. Results are displayed in Table 3 and explanation in text as follows:

Theoretically-guided intervention studies had higher ES, as did the ones involving community health workers and those conducted at broader level involving multiple locations within the country (Page 26, Lines 302-303)

6. Can the authors add some descriptions ahead (e.g., in the Method section) or somewhere appropriate about what the characteristic “Theoretically guided intervention” exactly indicate?

The detail has been added in the methods section as: 

… guidance by a theoretical or behaviour change model (theoretically guided) (Page 6, Lines 144-145)

7. Would there be any reason that the characteristic “Date of publication” divide into the three categories? It would be better for readers if clarifying and adding this on the manuscript.

There was no specific purpose, the date of publication was evenly distributed according to decades i.e., 1990-2000, 2001-2010, 2011-2021 

8. In Discussion, please give a more deep discussion on the following: (1) future research implication and direction based on this meta-analysis; and (2) relevant screening policies implications for immigrants.

Future research implication and direction, as suggested by Reviewer 1 also, has been added as follows:

None of the included studies reported cost-effectiveness of the intervention used, it is recommended to include this outcome in future studies. This would be of high practical value, allowing the policymakers to understand the value of the choice of intervention, including of HPV self-sampling method. A recent systematic review of studies assessing cost of HPV self-sampling compared to standard screening strategies, reported it to be highly cost -effective for under-screened women in high income countries, either when offered alone or in combination with other strategies[73] . Furthermore, effect of certain variables of interest, such as age and length of stay of participants in the country, could be explored through meta-regression in future studies. (Page 32, Lines 391-399)

Implications regarding relevant screening policies has been described as:

Therefore, policies designed to address cervical screening services and interventions for migrants should take into consideration, relevance of cultural responsiveness when including components such as information provision, navigation as well as financial incentives. (Page 29, Lines: 382-385).

Response to Editor Comments

The reference has not been included.

---

## [Decision Letter · Decision Letter 1]

18 Jan 2023

PONE-D-22-22346R1Interventions to increase cervical screening uptake among immigrant women: a systematic review and meta-analysisPLOS ONE

Dear Dr. Joanne Marie Cairns,

Thank you for submitting your manuscript to PLOS ONE. After careful consideration, we feel that it has merit but does not fully meet PLOS ONE’s publication criteria as it currently stands. Therefore, we invite you to submit a revised version of the manuscript that addresses the points raised during the review process.

We look forward to receiving your revised manuscript.

Kind regards,

Gulzhanat Aimagambetova

Academic Editor

PLOS ONE

Journal Requirements:

Reviewers' comments:

Reviewer's Responses to Questions

**Comments to the Author**

1. If the authors have adequately addressed your comments raised in a previous round of review and you feel that this manuscript is now acceptable for publication, you may indicate that here to bypass the “Comments to the Author” section, enter your conflict of interest statement in the “Confidential to Editor” section, and submit your "Accept" recommendation.

Reviewer #2: All comments have been addressed

Reviewer #3: (No Response)

2. Is the manuscript technically sound, and do the data support the conclusions?

Reviewer #2: Yes

Reviewer #3: Yes

3. Has the statistical analysis been performed appropriately and rigorously? 

Reviewer #2: Yes

Reviewer #3: Yes

4. Have the authors made all data underlying the findings in their manuscript fully available?

Reviewer #2: Yes

Reviewer #3: Yes

5. Is the manuscript presented in an intelligible fashion and written in standard English?

Reviewer #2: Yes

Reviewer #3: Yes

6. Review Comments to the Author

Reviewer #2: The manuscript has been well addressed in a quite comprehensive way and I have no further comments.

Reviewer #3: I read with great interest the Manuscript titled "Interventions to increase cervical screening uptake among immigrant women: a systematic review and meta-analysis " which falls within the aim of the Journal.

In my honest opinion, the topic is interesting enough to attract the readers’ attention. Methodology is accurate and conclusions are supported by the data analysis. Nevertheless, authors should clarify some point and improve the discussion citing relevant and novel key articles about the topic.

-The abstract should be improved (in particular the section on materials and methods) in order to better summarize the contents of the manuscript.

-The introduction should be extended and completed. I find interesting a reference to the efforts made for the prevention and early diagnosis of gynecological cancers (see PMID: 36141217).

- I also suggest authors to better organize the discussion section following this ideal structure: main findings of the study, strength and limitations of the study, implications and comparison with literature, future directions.

- Discussions can be expanded and improved by citing relevant articles (I suggest authors to read and insert in references the following article PMID: 35742340).

Considered all these points, I think it could be of interest for the readers and, in my opinion, it deserves the priority to be published after minor revisions.

7. PLOS authors have the option to publish the peer review history of their article (what does this mean?). If published, this will include your full peer review and any attached files.

Reviewer #2: No

Reviewer #3: **Yes: **Andrea Giannini

---

## [Author Response · Author response to Decision Letter 1]

24 Jan 2023

We are grateful to reviewer 3 for providing us feedback on the manuscript. We have addressed the comments and revised the manuscript as suggested. The detailed explanation to each comment raised by the reviewer is provided, the response is given (in blue) and changes/additions in the manuscript text (in red).

Reviewer #3: I read with great interest the Manuscript titled "Interventions to increase cervical screening uptake among immigrant women: a systematic review and meta-analysis " which falls within the aim of the Journal.

In my honest opinion, the topic is interesting enough to attract the readers’ attention. Methodology is accurate and conclusions are supported by the data analysis. Nevertheless, authors should clarify some point and improve the discussion citing relevant and novel key articles about the topic.

-The abstract should be improved (in particular the section on materials and methods) in order to better summarize the contents of the manuscript.

Abstract has been revised to include more details based on contents of the manuscript, keeping in view the abstract world limit. (Page 2, lines 19-44)

Numerous intervention studies have attempted to increase cervical screening uptake among immigrant women, nonetheless their screening participation remains low. This systematic review and meta-analysis aimed to summarise the evidence on interventions to improve cervical screening among immigrant women globally and identify their effectiveness. Databases PubMed, EMBASE, Scopus, PsycINFO, ERIC, CINAHL and CENTRAL were systematically searched from inception to October 12, 2021, for intervention studies, including randomised and clinical controlled trials (RCT, CCT) and one and two group pre-post studies. Peer-reviewed studies involving immigrant and refugee women, in community and clinical settings, were eligible. Comparator interventions were usual or minimal care or attention control. Data extraction, quality appraisal and risk of bias were assessed by two authors independently using COVIDENCE software. Narrative synthesis of findings was carried out, with the main outcome measure defined as the cervical screening uptake rate difference pre- and post-intervention followed by random effects meta-analysis of trials and two group pre-post studies, using Comprehensive Meta-Analysis software, to calculate pooled rate ratios and adjustment for publication bias, where found. The protocol followed PRISMA guidelines and was registered prospectively with PROSPERO (CRD42020192341). 1,900 studies were identified, of which 42 (21 RCTS, 4 CCTs, and 16 pre-post studies) with 44,224 participants, were included in the systematic review, and 28 with 35,495 participants in the meta-analysis. Overall, the uptake difference rate for interventions ranged from -6.7 to 96%. Meta-analysis demonstrated a pooled rate ratio of 1.15 (95% CI 1.03-1.29), with high heterogeneity. Culturally sensitive, multicomponent interventions, using different modes of information delivery and self-sampling modality were most promising. Interventions led to at least 15% increase in cervical screening participation among immigrant women. Interventions designed to overcome logistical barriers and use multiple channels to communicate culturally appropriate health promotion messages are most effective at achieving cervical screening uptake among immigrant women.

-The introduction should be extended and completed. I find interesting a reference to the efforts made for the prevention and early diagnosis of gynecological cancers (see PMID: 36141217).

Introduction has been revised, with addition of the suggested reference as follows (Page 3, Lines 48-50):

Research evidence clearly shows that secondary prevention in terms of screening can effectively reduce cervical cancer mortality [5]. Screening options now being employed worldwide include Pap and HPV test [6].

6. Giannini A, Bogani G, Vizza E, Chiantera V, Laganà AS, Muzii L, Salerno MG, Caserta D, D’Oria O. Advances on Prevention and Screening of Gynecologic Tumors: Are We Stepping Forward?. InHealthcare 2022 Aug 24 (Vol. 10, No. 9, p. 1605). MDPI.

- I also suggest authors to better organize the discussion section following this ideal structure: main findings of the study, strength and limitations of the study, implications and comparison with literature, future directions.

The discussion has been re organised following the suggested structure. The strengths and limitations have been moved before discussing implications and comparison with literature as follows (Page 27, Lines 296-307):

This study critically reviewed and meta-analysed interventions to increase cervical screening uptake among immigrant women globally. The review found that culturally appropriate interventions such as those providing HPV self-sampling methodology and targeted clinics for immigrants are most effective. Meta-analysis found that multicomponent interventions were more beneficial than single component ones, as were those theoretically guided, delivered in-person and using multiple formats of information delivery. Participant characteristics, such as baseline screening status also influenced the success of the interventions, as did the type of intervention selected for the control group.

To our knowledge this is the first systematic review and meta-analysis to comprehensively map the global evidence on the effectiveness of interventions to increase cervical screening in immigrant women. It compared the intervention effect sizes based on characteristics such as delivery format, involvement of HCPs, modes of educative material, screening status of participants and type of control group. The strengths of this study include following a prospectively registered protocol, clearly and transparently outlining our search strategy and methods of analysis, having multiple reviewers independently working on each review stage, and investigating novel methods of encouraging screening i.e., self-sampling, not included in previous intervention reviews on immigrants. However, this review also has certain limitations. First, although we conducted a comprehensive search of multiple databases, some studies might not have been located. We tried to overcome this through hand citation searches. Second, we found the EPHPP tool was more favorable towards experimental studies compared to non-controlled studies resulting in most studies being scored of weak quality. We also found significant heterogeneity in the studies limiting the number we could include in the sub-group meta-analyses.

The meta-analysis results suggested a low overall ES of 1.15 (95% CI 1.03–1.29) across all intervention studies after adjustment for publication bias…….

- Discussions can be expanded and improved by citing relevant articles (I suggest authors to read and insert in references the following article PMID: 35742340).

Considered all these points, I think it could be of interest for the readers and, in my opinion, it deserves the priority to be published after minor revisions.

We feel that the suggested reference fits better in the introduction section due to its focus on treatment of early and late stages of cervical cancer, rather than in the discussion. Therefore, we have included it in the introduction as follows (Page 3, lines 50-51):

Advances in biomedical research has led to introduction of novel surgical, radiotherapeutic and systemic options for the treatment of cervical cancer [4].

4. D’Oria O, Corrado G, Laganà AS, Chiantera V, Vizza E, Giannini A. New Advances in Cervical Cancer: From Bench to Bedside. International Journal of Environmental Research and Public Health. 2022 Jun 9;19(12):7094.

---

## [Decision Letter · Decision Letter 2]

7 Feb 2023

Interventions to increase cervical screening uptake among immigrant women: a systematic review and meta-analysis

PONE-D-22-22346R2

Dear Dr. Joanne Marie Cairns,

We’re pleased to inform you that your manuscript has been judged scientifically suitable for publication and will be formally accepted for publication once it meets all outstanding technical requirements.

Kind regards,

Gulzhanat Aimagambetova

Academic Editor

PLOS ONE

Additional Editor Comments (optional):

Reviewers' comments:

Reviewer's Responses to Questions

**Comments to the Author**

1. If the authors have adequately addressed your comments raised in a previous round of review and you feel that this manuscript is now acceptable for publication, you may indicate that here to bypass the “Comments to the Author” section, enter your conflict of interest statement in the “Confidential to Editor” section, and submit your "Accept" recommendation.

Reviewer #2: All comments have been addressed

Reviewer #3: All comments have been addressed

2. Is the manuscript technically sound, and do the data support the conclusions?

Reviewer #2: Yes

Reviewer #3: Yes

3. Has the statistical analysis been performed appropriately and rigorously? 

Reviewer #2: Yes

Reviewer #3: Yes

4. Have the authors made all data underlying the findings in their manuscript fully available?

Reviewer #2: Yes

Reviewer #3: Yes

5. Is the manuscript presented in an intelligible fashion and written in standard English?

Reviewer #2: Yes

Reviewer #3: Yes

6. Review Comments to the Author

Reviewer #2: The manuscript has been addressed well and with a certain level of quality to me. I have no further comments.

Reviewer #3: Dear authors, thank you for sending me the correct manuscript.

I read your work with great interest and pleasure. The work with the changes made after my advice and those of the other authors is complete and, in my opinion, ready for publication.

7. PLOS authors have the option to publish the peer review history of their article (what does this mean?). If published, this will include your full peer review and any attached files.

Reviewer #2: No

Reviewer #3: **Yes: **Andrea Giannini

---

## [Editor Report · Acceptance letter]

22 May 2023

PONE-D-22-22346R2 

Interventions to increase cervical screening uptake among immigrant women: a systematic review and meta-analysis 

Dear Dr. Cairns:

I'm pleased to inform you that your manuscript has been deemed suitable for publication in PLOS ONE. Congratulations! Your manuscript is now with our production department. 

Kind regards, 

on behalf of

Dr. Gulzhanat Aimagambetova 

Academic Editor

PLOS ONE